# A model of persistent breaking of continuous symmetry

**Noam Chai,**[1] **Anatoly Dymarsky,**[2,3] **Mikhail Goykhman,**[1,4] **Ritam Sinha,**[1] **and Michael Smolkin**[1]

[1] *The Racah Institute of Physics, The Hebrew University of Jerusalem,*
*Jerusalem 91904, Israel*

[2] *Department of Physics and Astronomy,*
*University of Kentucky, Lexington, KY 40506*

[3] *Skolkovo Institute of Science and Technology,*
*Skolkovo Innovation Center, Moscow, Russia, 143026*

[4] *William I. Fine Theoretical Physics Institute, University of Minnesota, Minneapolis, MN 55455,*
*USA*
*E-mail:* noam.chai@mail.huji.ac.il, a.dymarsky@uky.edu,
goykhman@umn.edu, ritam.sinha@mail.huji.ac.il,
michael.smolkin@mail.huji.ac.il

ABSTRACT: We consider a UV-complete field-theoretic model in general dimensions, including $d = 2 + 1$, that exhibits spontaneous breaking of a continuous symmetry, persisting to arbitrarily large temperatures. Our model consists of two copies of the long-range vector models, with $O(m)$ and $O(N - m)$ global symmetry groups, perturbed by double-trace operators. Using conformal perturbation theory we find weakly-coupled IR fixed points for $N \geq 6$ that reveal a spontaneous breaking of global symmetry. Namely, at finite temperature the lower rank group is broken, with the pattern persisting at all temperatures due to scale-invariance. We provide evidence that the models in question are unitary and invariant under full conformal symmetry. Furthermore, we show that this model exhibits a continuous family of weakly interacting field theories at finite $N$.

# 1 Introduction

The phenomenon of persistent symmetry breaking (PSB), *i.e.*, spontaneous breaking of a global symmetry that persists at arbitrarily high temperatures, was first noticed by Weinberg [1]. Since then it has been actively discussed in cosmology [2–7], quantum field theory [8–16] and holography [17–24].

The AdS/CFT candidates for persistent order are perturbatively stable, but the symmetric phase has smaller free energy. In fact, there are a number of theoretical results, which guarantee symmetry restoration at sufficiently high temperatures in certain particular settings, yet in some models the PSB behavior is possible. Thus, during the last year a number of papers studied critical points of the bi-conical $O(m) \times O(N - m)$ model in the context of the PSB. At finite temperature $T$, these models exhibit spontaneous symmetry breaking, and because of scale-invariance symmetry breaking persists to an arbitrarily high $T$ [8, 11–13]. Furthermore, the authors of [14, 16] constructed critical gauge theory models in $3 + 1$ dimensions which exhibit symmetry breaking at arbitrary high temperatures in the infinite $N$ limit. Note that starting from a scale invariant model resolves the issue of UV completeness, present in the original example of PSB [1]. There are also examples of non-unitary models of persistent breaking in the presence of chemical potential [9, 10, 25–27].[1]

The aforementioned works suggest that persistent breaking of global symmetries is possible. Therefore, the standard logic which suggests that the disordered phase has larger entropy $\mathcal{S}$, and therefore for sufficiently large $T$ the disordered phase would have smallest free energy $\mathcal{F} = \mathcal{E} - T\mathcal{S}$, under certain circumstances can break down.

---

[1]Persistent order can be also considered in the context of spontaneous breaking of higher-form symmetries, see *e.g.,* [28]. In this work we focus on the ordinary (zero-form) symmetries.

In this paper, we study persistent symmetry breaking in critical models with the long-range interactions. While the models of interest can be defined in general dimensions $1 < d < 4$, our primary interest will be in physically motivated case of $d = 2 + 1$. In $2 + 1$ dimensions, the Coleman-Hohenberg-Mermin-Wagner (CHMW) theorem prohibits spontaneous breaking of continuous symmetries at non-zero temperature [29–31]. However, the necessary assumptions for the CHMW result can be evaded, for instance, by focusing on spontaneous breaking of discrete symmetries. This is what was done in [8], which studied persistent breaking of the global $\mathbb{Z}_2$ symmetry in the $\mathbb{Z}_2 \times O(N)$ long-range vector model. Moreover, the continuous symmetry in similar models can also be broken persistently in $d = 2 + 1$, owing to yet another way around the assumptions behind the CHMW theorem: by considering systems with long-range interactions [30, 32, 33].[2] In this paper the bi-conical long-range critical $O(m) \times O(N - m)$ vector model is studied perturbatively, close to the Gaussian mean field theory point. We explore IR flows and properties of the IR fixed points at zero temperature. As a result interesting in its own right, we show that our model in $3d$ exhibits a conformal manifold of interacting fixed points at infinite $N$. At finite $N$ the continuous space of interacting fixed points persists and is parametrized by a continuous family of Gaussian theories in the UV. While the continuous space of CFTs by itself is not uncommon, to our knowledge there were no prior known examples of the non-SUSY finite $N$ models admitting an exactly marginal deformation.

One of the main advantages of considering long-range interactions is the possibility to formulate a model admitting perturbative treatment in arbitrary number of dimensions $d$. This is achieved by the choice of the critical exponent of the bi-local kinetic term of the generalized free field. Subsequently turning on a quartic coupling results in an RG flow, terminating at a weakly-coupled IR fixed point, located in the perturbative vicinity of the long-range mean field theory point [35]. While this approach bears certain similarity to the Wilson-Fisher $\epsilon$-expansion [36], it is applicable in any $d$, including $d = 3$.

An important feature of the long-range models is absence of the local stress tensor. Without the stress tensor full conformal symmetry of the scale-invariant fixed point is not manifest. An extensive supporting argument in favor of the full conformal invariance of the RG fixed point of the long-range Ising model (i.e., $O(N)$ vector model with $N = 1$) has recently been put forward in [37–40]. For large $N$, the long-range $O(N)$ vector model at criticality has been studied in [41, 42] using $1/N$ expansion, with the most recent works [43, 44] providing a strong evidence that the critical regime of this model is in fact described by a CFT (see also [41, 42, 45–49] for previous calculations of critical exponents).[3] At the same time, full conformal invariance of the critical bi-conical long-range $O(m) \times O(N - m)$ model is less understood. Full conformal symmetry, if present, would restrict the functional

---

[2]This is to be contrasted with the earlier models of PSB [11–13], that studied the local bi-conical $O(m) \times O(N - m)$ vector models, where continuous symmetry breaking can occur only in non-integer dimensions $3 < d < 4$. The latter choice of dimension, however, raises the issue of unitarity [34].

[3] It was argued in [38, 39] that the non-local CFT describing the IR fixed point of the long-range $\phi^4$ model can also be found at the IR end-point of an RG flow triggered by coupling the short-range vector field to the generalized free field of dimension $(d + s)/2$. Such a coupling is irrelevant in the short-range regime, and can be studied perturbatively near the long-range to short-range crossover point. $O(N)$ generalization of this IR duality has recently been discussed in [43].

form of the three-point functions of primary operators, and ensure that cross-correlators of primaries with different conformal dimensions vanish. We perform several consistency checks of the long-range $O(m) \times O(N-m)$ vector models at criticality and confirm expected behavior of the two and three-point functions, dictated by full conformal symmetry. In the process, we verify that the anomalous dimensions of all considered single-trace and double-trace operators remain real, which is a necessary condition of unitarity.

This paper is organized as follows. In section 2.1 we begin by defining our model. Working at the linear order in perturbation theory near free critical point, we derive the RG flow equations for the quartic double-trace interaction coupling constants. Then we analyze the fixed points of the RG flow for different choices of scaling dimensions of the scalar fields vector multiplets $\phi_{1,2}$. In particular, we focus on the models admitting negative fixed point value of the coupling constant $g_3$ corresponding to the quartic operator $\phi_1^2 \phi_2^2$. Behavior of such fixed points at finite temperature is then explored in section 3. Specifically, we demonstrate that some of the fixed points $g_3 < 0$ lead to an instability of the symmetric vacuum $\phi_1 = 0$ of the effective action at $T > 0$. The resulting model therefore breaks $O(m)$ symmetry spontaneously at any non-zero temperature, exhibiting the phenomenon of PSB.

The model in question admits a continuous family of interacting fixed points, that we discuss in section 2.2. We additionally explore the nature of the critical regime of the considered bi-conical model at zero temperature in section 2.3, where we calculate anomalous dimensions of various single-trace (quadratic) and double-trace (quartic) operators. To this end, we take into account the operator mixing effect, and diagonalize the correlation matrices by finding the true basis of primary operators. As we mentioned above, fixed point of an interacting long-range model, lacking a local stress-energy tensor, might end up being scale-invariant but not conformal invariant. We carry out several checks of the full conformal symmetry in section 2.4. In that section, we calculate cross-correlator between quadratic and quartic conformal primaries, and demonstrate that they vanish at the considered order in $\epsilon$-expansion. We also calculate three-point correlator and show that at leading order it agrees with the form dictated by the conformal symmetry.

The results of the paper are summarized in section 4.

## 2 Long range $O(m) \times O(N-m)$ vector model at criticality

### 2.1 The model and the RG flow

Consider the following Gaussian action in $1 \leq d < 4$ dimensions,

$$S_0 = \mathcal{N}_1 \int d^d x_1 \int d^d x_2 \frac{\vec{\phi}_1(x_1) \cdot \vec{\phi}_1(x_2)}{|x_1 - x_2|^{2(d-\Delta_{\phi_1})}} + \mathcal{N}_2 \int d^d x_1 \int d^d x_2 \frac{\vec{\phi}_2(x_1) \cdot \vec{\phi}_2(x_2)}{|x_1 - x_2|^{2(d-\Delta_{\phi_2})}} . \quad (2.1)$$

The model (2.1) describes two real-valued generalized free scalar fields $\vec{\phi}_1$ and $\vec{\phi}_2$ transforming in vector representation of the $O(m)$ and $O(N-m)$ global symmetry groups. Our conventions are such that $m < N$. The coefficients $\mathcal{N}_{1,2}$ are fixed so that the two point functions of $\vec{\phi}_{1,2}$ in position space are normalized to one. The scaling dimensions of the

generalized free fields are

$$\Delta_{\phi_i} = \frac{d - \epsilon_i}{4}, \quad i = 1, 2 . \tag{2.2}$$

For brevity, we will be suppressing $O(m)$, $O(N - m)$ vector indices below.

In what follows, we are going to consider deformation of the free action (2.1) by the following double-trace operators

$$\mathcal{O}_1 = (\phi_1^2)^2 , \mathcal{O}_2 = (\phi_2^2)^2, \; \mathcal{O}_3 = \phi_1^2\phi_2^2 . \tag{2.3}$$

Choosing $\epsilon_i \ll 1$, one can make these operators weakly relevant, with the leading order scaling dimensions $\Delta_1 = 4\Delta_{\phi_1}$, $\Delta_2 = 4\Delta_{\phi_2}$ and $\Delta_3 = 2(\Delta_{\phi_1} + \Delta_{\phi_2})$. At the same time, the leading-order two-point functions of these operators are given by

$$\langle \mathcal{O}_i(x)\mathcal{O}_j(0) \rangle = \delta_{ij} \frac{N_i}{|x|^{2\Delta_i}},$$

$$N_1 = 8m^2\Big(1 + \frac{2}{m}\Big), \; N_2 = 8(N - m)^2\Big(1 + \frac{2}{N - m}\Big), \; N_3 = 4m(N - m) . \tag{2.4}$$

Similarly, the leading-order three-point functions

$$\langle \mathcal{O}_i(x_1)\mathcal{O}_j(x_2)\mathcal{O}_k(x_3) \rangle = \frac{C_{ij}^k N_k}{|x_{12}|^{\Delta - 2\Delta_k}|x_{23}|^{\Delta - 2\Delta_i}|x_{13}|^{\Delta - 2\Delta_j}},$$

$$\Delta = \Delta_i + \Delta_j + \Delta_k, \tag{2.5}$$

are fixed by the OPE coefficients (we list only the non-zero ones)

$$C_{11}^1 = 8\,(m + 8) \;,\; C_{33}^1 = 2(N - m) \;,\; C_{13}^3 = 4\,(m + 2) \;,\; C_{33}^3 = 16 \;,$$
$$C_{32}^3 = 4(N - m + 2) \;,\; C_{33}^2 = 2m \;,\; C_{22}^2 = 8\,(N - m + 8) \;. \tag{2.6}$$

The latter are related by

$$C_{ij}^k = C_{ik}^j N_j / N_k. \tag{2.7}$$

Consider now the following deformation of the Gaussian theory (2.1)

$$S = S_0 + \sum_{i=1}^3 \frac{g_i\mu^{\epsilon_i}}{N} \int d^d x \, \mathcal{O}_i(x) , \tag{2.8}$$

where $\mu$ is an arbitrary RG scale, and we also denoted $\epsilon_3 = (\epsilon_1 + \epsilon_2)/2$. This deformation induces an RG flow, that at the next-to-leading (one-loop) order in perturbation around the free regime has the form

$$\mu\frac{dg_i}{d\mu} = -\epsilon_i g_i + \frac{\pi^{d/2}}{N\Gamma\left(\frac{d}{2}\right)} \sum_{j,k} C_{jk}^i g_j g_k + \mathcal{O}(g_i^3) . \tag{2.9}$$

We are interested in the interacting IR critical regime of the model (2.8). To this end, we need to determine fixed-points of the RG flow (2.9). At the one-loop order, the critical parameters can be found by plugging in the values of the OPE coefficients (2.6) into the r.h.s. of (2.9). One of the fixed points, with $g_3 = 0$, describes two decoupled copies of the

long-range vector models. We will not be considering this fixed point it in what follows. When $g_3 \neq 0$ the fixed points can be found by solving the following system of coupled second-order equations,

$$\tilde{g}_1 = \frac{C_{11}^1}{N} \, \tilde{g}_1^2 + \frac{C_{33}^1}{N} \Big( \frac{\alpha+1}{2\alpha} \Big)^2 \tilde{g}_3^2 \, ,$$
$$\tilde{g}_2 = \frac{C_{22}^2}{N} \, \tilde{g}_2^2 + \frac{C_{33}^2}{N} \Big( \frac{\alpha+1}{2} \Big)^2 \tilde{g}_3^2 \, , \tag{2.10}$$
$$1 = \frac{C_{13}^3}{N} \frac{4\alpha}{\alpha+1} \, \tilde{g}_1 + \frac{C_{33}^3}{N} \tilde{g}_3 + \frac{C_{32}^3}{N} \frac{4}{\alpha+1} \, \tilde{g}_2 \, .$$

Here we have rescaled the couplings as $g_i = \tilde{g}_i \frac{\Gamma\left(\frac{d}{2}\right)}{\pi^{d/2}} \epsilon_i$, and defined the parameter

$$\alpha = \frac{\epsilon_1}{\epsilon_2} \, . \tag{2.11}$$

Notice that the equations (2.10) are invariant under the redefinitions $\alpha \to 1/\alpha$, $m \to N-m$, $\tilde{g}_1 \leftrightarrow \tilde{g}_2$, originating from a simple interchange of notations for the fields $\phi_{1,2}$ in the original action (2.8). Therefore there is a one-to-one correspondence between the family of solutions with $\alpha \geq 1$ and $\alpha \leq 1$. Without loss of generality we restrict our analysis to the case $\alpha \geq 1$.

It can be easily seen, as we discuss below in section 3, that the persistent symmetry breaking is only possible when the fixed-point value of the coupling constant $g_3$ is negative. These are the solutions of (2.10) that we wish to explore.

While in general it is difficult to solve the system of second-order equations (2.10) analytically, a simplification can be achieved in the large $N$ limit. We begin by considering the case of $N \gg 1$, $m \gg 1$, with fixed $m/N = \mathcal{O}(1)$. Denoting

$$x_1 = \frac{m}{N} \, , \qquad x_2 = 1 - x_1 \, , \tag{2.12}$$

we can re-write equations (2.10) at the leading order in $1/N$ as

$$\tilde{g}_1 = 8x_1 \, \tilde{g}_1^2 + 2x_2 \Big( \frac{\alpha+1}{2\alpha} \Big)^2 \tilde{g}_3^2 \, ,$$
$$\tilde{g}_2 = 8x_2 \, \tilde{g}_2^2 + 2x_1 \Big( \frac{\alpha+1}{2} \Big)^2 \tilde{g}_3^2 \, , \tag{2.13}$$
$$1 = \frac{16}{\alpha+1} \left( \alpha \, x_1 \, \tilde{g}_1 + x_2 \, \tilde{g}_2 \right) \, .$$

These equations admit a solution only if $\alpha = 1$, in which case they are degenerate, and a line of fixed points consisting of two branches emerges,[4]

$$\tilde{g}_1^\pm = \frac{1 \pm \sqrt{1 - 64x_1 x_2 \tilde{g}_3^2}}{16x_1} \, ,$$
$$\tilde{g}_2^\pm = \frac{1 \mp \sqrt{1 - 64x_1 x_2 \tilde{g}_3^2}}{16x_2} \, , \tag{2.14}$$
$$\tilde{g}_3 \in \Big[ -\frac{1}{8\sqrt{x_1 x_2}} \, , \, +\frac{1}{8\sqrt{x_1 x_2}} \Big].$$

---

[4] This solution was in fact first found in [13], that considered the bi-conical $O(m) \times O(N-m)$ model in $4 - \epsilon$ dimensions. At the one-loop order in perturbation theory, the fixed point of this model is determined by the equations (2.10) with $\alpha = 1$.

The two branches meet at the end points $\tilde{g}_3 = \pm 1/(8\sqrt{x_1 x_2})$, forming a circle, see Fig. 1. The coordinate on this circle parametrizes one of the directions on the two-dimensional conformal manifold of the model (2.8); we will discuss the latter in more detail in section 2.2. The corresponding exactly marginal operator $\mathcal{O}'_+$, obtained by mixing the double-trace operators $\mathcal{O}_i$, $i = 1, 2, 3$, will be derived below in section 2.3.

There is a special point $\tilde{g}_1 = \tilde{g}_2 = 1/8$, $\tilde{g}_3 = 1/4$ where $O(m) \times O(N-m)$ symmetry is enhanced to the full $O(N)$. Being independent of $x_1$, it can be found as the intersection point of curves with different $x_1$. Since the RG equations in the infinite $N$ limit, given by (2.13), are invariant w.r.t. $\tilde{g}_3 \to -\tilde{g}_3$, another intersection point is given by $\tilde{g}_1 = \tilde{g}_2 = 1/8$, $\tilde{g}_3 = -1/4$.

Additionally, each curve has "decoupled" points $\tilde{g}_1^- = \tilde{g}_3^- = 0, \tilde{g}_2^- = \frac{1}{8x_2}$ and $\tilde{g}_2^+ = \tilde{g}_3^+ = 0, \tilde{g}_1^+ = \frac{1}{8x_1}$ where only one critical long-range vector model survives.

To find corrections to the large-$N$ solution (2.14) at the next-to-leading order in $1/N$ expansion, we substitute the ansatz

$$\tilde{g}_i \to \tilde{g}_i + \delta\tilde{g}_i/N + \mathcal{O}(1/N^2), \quad \alpha = 1 + \delta\alpha/N + \mathcal{O}(1/N^2) \tag{2.15}$$

into (2.10), and linearize over the $1/N$ terms. This yields

$$\begin{pmatrix} 1 - 16\tilde{g}_1 x_1 & 0 & -4\tilde{g}_3 x_2 \\ 0 & 1 - 16\tilde{g}_2 x_2 & -4\tilde{g}_3 x_1 \\ 2x_1 & 2x_2 & 0 \end{pmatrix} \begin{pmatrix} \delta\tilde{g}_1 \\ \delta\tilde{g}_2 \\ \delta\tilde{g}_3 \end{pmatrix} = \begin{pmatrix} 64\tilde{g}_1^2 \\ 64\tilde{g}_2^2 \\ -4\sum_{i=1}^3 \tilde{g}_i \end{pmatrix} + \begin{pmatrix} -2\tilde{g}_3^2 x_2 \\ 2\tilde{g}_3^2 x_1 \\ \tilde{g}_2 - (\tilde{g}_1 + \tilde{g}_2)x_1 \end{pmatrix} \delta\alpha. \tag{2.16}$$

where we suppressed the $\mathcal{O}(1/N^2)$ terms. The matrix on the left hand side is singular and the linear system has a solution if and only if the following constraint is satisfied:

$$4\left[3 - 8\tilde{g}_2(3 - 16\tilde{g}_3 x_1 x_2) - 8\tilde{g}_3 x_1\left(1 + 4\tilde{g}_3(1 - 2x_1)\right)\right] - x_1\,\delta\alpha = 0 . \tag{2.17}$$

This means every fixed point of the infinite $N$ conformal manifold (2.14) survives finite $N$ corrections in a theory with the appropriate chosen $\alpha$.

Let us demand that finite but large $N$ theory has $\alpha = 1$. Corresponding fixed points are the intersections of (2.14) with the following surface, see Fig. 1,

$$\frac{3}{8} = \tilde{g}_2(3 - 16\tilde{g}_3 x_1 x_2) + \tilde{g}_3 x_1\left(1 + 4\tilde{g}_3(1 - 2x_1)\right) . \tag{2.18}$$

One of the intersection points is $\tilde{g}_{1,2} = 1/8, \tilde{g}_3 = 1/4$, which is the theory with full $O(N)$ symmetry. Another fixed point has $g_3 < 0$, and exhibits persistent symmetry breaking, as discussed later in section 3. Similar conclusion holds for other values of $\alpha$.

Another class of tractable large-$N$ CFTs can be obtained by considering the limit $N \to \infty$ with fixed $m$. In this case coupling constants will grow linearly with $N$, $\tilde{g}_i = \tilde{g}_i^{(1)}N + \tilde{g}_i^{(0)} + \dots$. Substituting this into (2.10) and solving order by order in the large $N$ limit yields a real

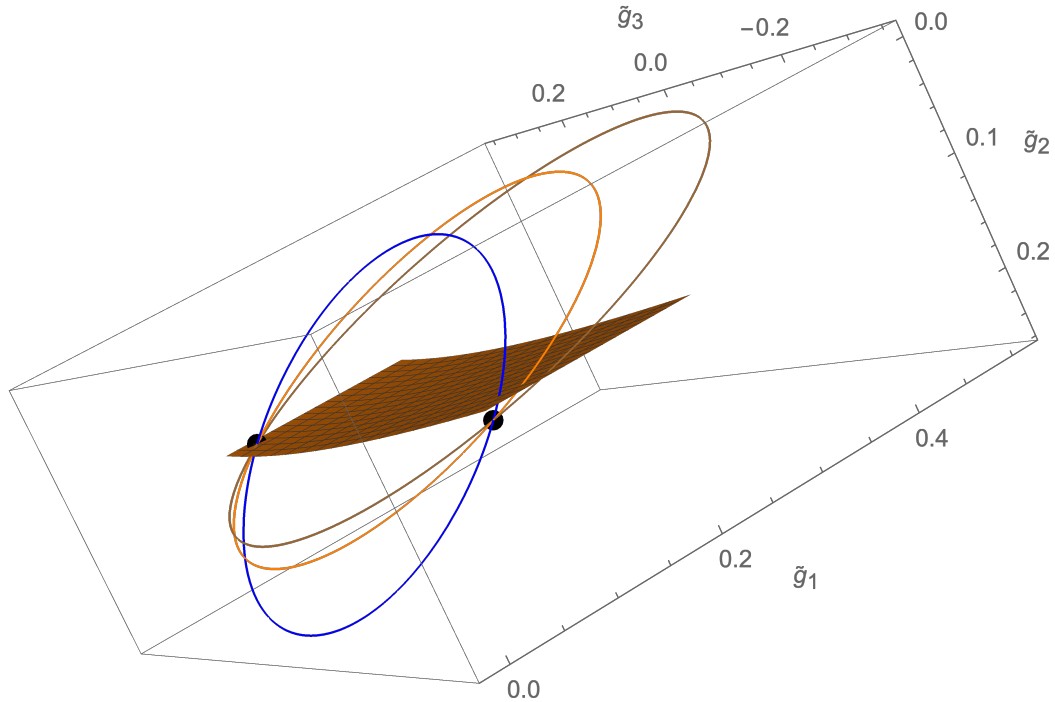

**Figure 1**. Conformal manifolds (2.14) for $x_1 = 1/2$ (blue), $x_1 = 1/3$ (orange) and $x_1 = 1/4$ (brown). Black points correspond to $\tilde{g}_1 = \tilde{g}_2 = 1/8$, $\tilde{g}_3 = \pm 1/4$ which are common for all $x_1$. Orange surface, (2.18) with $x_1 = 1/3$, intersects the orange curve at two points (with positive and negative $\tilde{g}_3$) which are two theories with $m/N = 1/3$, $\alpha = 1$, and large but finite $N$. Theory with positive $\tilde{g}_3 = 1/4$ has the global symmetry enhanced to $O(N)$, while the theory with negative $\tilde{g}_3 = -1/4$ at finite temperatures exhibits persistent symmetry breaking $O(m) \times O(N-m) \to O(m-1) \times O(N-m)$.

solution of the form[5]

$$
\tilde{g}_1 = \frac{N}{8(m+8)} - 2\Big(\frac{g_3^{(0)}(m+2)}{m+8}\Big)^2 , \quad \tilde{g}_2 = 0 , \quad \tilde{g}_3 = g_3^{(0)} , \quad \alpha = \frac{m+8}{m-4} , \ m > 4 ,
$$

$$
\tilde{g}_1 = \frac{N}{8(m+8)} - 72\Big(\frac{g_3^{(0)}}{m+8}\Big)^2 , \quad \tilde{g}_2 = \frac{1}{8} , \quad \tilde{g}_3 = g_3^{(0)} , \quad \alpha = -\frac{m+8}{m-4} , \ m < 4 ,
$$

(2.19)

where the $\mathcal{O}(1/N)$ terms are suppressed. The solution is not unique, the coefficient $g_3^{(0)}$ cannot be fixed without accounting for higher order corrections in $1/N$. Moreover, $\alpha$ is constrained,

$$
\alpha = \alpha_c \equiv \Big| \frac{m+8}{m-4} \Big| .
$$

(2.20)

Higher order in $1/N$ contributions remove the ambiguity in (2.19). Furthermore, higher order terms decrease the value of $\alpha$ such that $1 \leq \alpha \leq \alpha_c$, *i.e.*, $\alpha_c$ is an upper critical value above which only complex fixed points exist. We verified explicitly there are fixed points

---

[5]As expected, in the limit $x_1 = m/N \to 0$ the fixed points $(\tilde{g}_1^+, \tilde{g}_2^+, \tilde{g}_3)$ of (2.14) converge to (2.19) for $m \gg 1$.

with large and fixed $m$, $N \to \infty$, $\alpha \leq \alpha_c$ which have $\tilde{g}_3 < 0$ and which exhibit persistent symmetry breaking.

Fixed points for theories with small $N \sim \mathcal{O}(1)$ are difficult to analyze analytically but (2.10) can be readily solved numerically. Ultimately, one finds a large family of stable fixed points with real-valued couplings and $\tilde{g}_3 < 0$. For $m = 1$ there are such fixed points for any $N \geq 6$ and appropriate $\alpha$. Thus, persistent breaking of discrete global symmetry is possibly already in the $\mathbb{Z}_2 \times O(5)$ model. In [8] it was reported that $N > 17$ is necessary for symmetry breaking because the analysis there was restricted to $\alpha = 1$ theories, while here we consider more general models with arbitrary positive $\alpha$. For $m = 2$ fixed points with $\tilde{g}_3 < 0$ (and persistent breaking) appear for all $N \geq 7$ and the appropriate $\alpha$. More generally for $m \leq 5$ we find $\tilde{g}_3 < 0$ fixed points for all $N \geq m + 5$ and for $m \geq 5$ for all $N \geq 2m$, using the nomenclature $m \leq N - m$.

We note that negative $\tilde{g}_3$ is necessary but not sufficient for persistent symmetry breaking. So far our emphasis was on the value of $\tilde{g}_3$. We will consider finite temperature effects in more detail in section 3.

## 2.2 Family of interacting CFTs at finite $N$

In section 2.1, while working in the infinite $N$ limit, we encountered a conformal manifold of interacting fixed points. In this section, we are going to discuss the origin of this conformal manifold, as well as its fate at finite $N$.

We are interested in analyzing fixed points of the model (2.8). At the Gaussian fixed point, $g_i = 0$, $i = 1, 2, 3$, we have a continuous family of CFTs, parametrized by scaling dimensions $\Delta_{\phi_j}$, $j = 1, 2$. In the previous section, we expressed these scaling dimensions as $\Delta_{\phi_j} = (d - \epsilon_j)/4$, in terms of the ratio $\alpha = \epsilon_1/\epsilon_2$, and $\epsilon_2$. This is convenient to do because in interacting theory, while working at the linear order in perturbation theory (in $\epsilon_{1,2}$) one can simply factor $\epsilon_2$ out of all expressions.

Let us fix $\epsilon_i$ in the UV. Since (2.1) are non-local, they are not getting renormalized, and therefore values of $\Delta_{\phi_1}, \Delta_{\phi_2}$ remain the same along the RG flow. Now let us assume this theory admits a non-trivial interacting fixed point $(g_1^\star, g_2^\star, g_3^\star)$. There are different scenarios concerning whether it belongs to a continuous family:

- Fixed point $(g_1^\star, g_2^\star, g_3^\star)$ is a solution of equations $\beta_i = 0$, $i = 1, 2, 3$, and it exists for some open set of $\Delta_{\phi_j}$, $j = 1, 2$. Then corresponding interacting fixed points form a continuous family, parametrized by $\Delta_{\phi_j}$, $j = 1, 2$, just like it was for the Gaussian fixed points. This is the general case scenario, which assumes the fixed point equations $\beta_i(g_j, \epsilon_k) = 0$ are non-degenerate.

- Fixed point $(g_1^\star, g_2^\star, g_3^\star)$ is a solution of $\beta_i = 0$, that exists only for the isolated point(s) on the $(\Delta_{\phi_1}, \Delta_{\phi_2})$ plane. There is no continuous family of interacting fixed points in this case.

- The system of equations $\beta_i = 0$, is degenerate w.r.t. $g_{1,2,3}$, and for consistency $\Delta_{\phi_1}, \Delta_{\phi_2}$ must be related to each other, forming a curve on the $(\Delta_{\phi_1}, \Delta_{\phi_2})$ plane.

There is a continuous family of interacting fixed points in this case, it is two-dimensional, and intersects the $(\Delta_{\phi_1}, \Delta_{\phi_2})$ plane along a one-dimensional curve.

In each scenario above, besides $\beta_i(g^*) = 0$, we also assumed $g_i^*$ satisfy additional constraints, e.g. $g_{1,2}^* > 0$, to ensure stability of the model.

In section 2.1, we saw that in the infinite $N$ limit we have the third scenario, while finite $N$ models follow first scenario, i.e. also admit a continuous family of fixed points. Specifically, in the infinite $N$ limit, for fixed $m/N = \mathcal{O}(1)$, we saw that the system of quadratic equations (2.13) is degenerate. Its admits a one-parametric family of solutions (2.14). The corresponding exactly marginal operator $\mathcal{O}'_+$, that we discuss in more detail latter in section 2.3, manifests existence of a one-dimensional conformal manifold. Together with the parameter $\epsilon_1 = \epsilon_2$, and associated non-local deformation of (2.1), we obtain a two-dimensional continuous family of CFTs. It is algebraically straightforward to find $1/N$ corrections to the leading order solution (2.14). For instance, at the next-to-leading order one needs to solve the linearized system of equations (2.16). The one-dimensional conformal manifold in the $\epsilon_1 = \epsilon_2$ theory, that we observed in the infinite $N$ limit, is lifted by the $1/N$ corrections: there is no marginal operator $\mathcal{O}'$ (for any possible mixing of the double-trace operators $\mathcal{O}_i$) in this case. However, imposing the constraint (2.17) one is able to see that by changing the parameters $\epsilon_{1,2}$ in the UV the two-parametric family of interacting CFTs survives. A similar argument will remain true at any order in $1/N$ expansion.

In fact, it is easy to see numerically that the two-parametric family of interacting CFTs exists at finite $N$. For instance, in Fig. 4 for the theory with $m = 1$, $N = 6$ we plot critical $\tilde{g}_3^*$ for various admissible values of $\alpha$ for which interacting fixed point exists. Analogous numerical analysis can be used to establish existence of the family of interacting CFTs for general $m, N$.[6]

## 2.3 Anomalous dimensions

Before we proceed with the finite temperature analysis in section 3 we would like to make sure the fixed points with negative $g_3$, which we found in section 2.1, correspond to UV complete unitary theories. For that purpose in this section we calculate anomalous dimensions of the composite operators $\mathcal{O}_i$[7]. We find the anomalous dimensions to be real, which is a necessary condition for unitarity.

We begin with repeating the derivation of the anomalous dimensions at leading order of the conformal perturbation theory, keeping in mind that our theory is non-local. Nevertheless it does not lead to any deviations from the standard result, as we see below. At

---

[6] Interestingly enough, there is the aforementioned family for $m = 1$ and any $N \geq 2$, except for $N = 4$, when the continuous family of solutions disappears. Solutions still seem to exist for a certain discrete set of $\alpha$, realizing second scenario above (although $\epsilon_1$ is still continuous). For example, for $\alpha = 1$, $N = 4$, $m \in \{1, 2, 3\}$, we obtain $\tilde{g}_1 = \tilde{g}_2 = \frac{1}{24}$, $\tilde{g}_3 = \frac{1}{12}$. This solution, however, has positive $g_3$ and hence does not exhibit the PSB.

[7] In this paper, we denote quartic and quadratic operators in the action as $\mathcal{O}_i$ and $\tilde{\mathcal{O}}_i$ respectively, while the analogous operators at the interacting fixed points are denoted with primes.

linear order the conformal perturbation theory gives

$$\langle \mathcal{O}_i(x_1)\mathcal{O}_j(x_2)\rangle = \delta_{ij}\frac{N_i}{|x|^{2\Delta_i}} - \sum_{k=1}^{3}\frac{g_k\mu^{\epsilon_k}}{N}\int d^dx_3\langle \mathcal{O}_i(x_1)\mathcal{O}_j(x_2)\mathcal{O}_k(x_3)\rangle + \mathcal{O}(g_k^2) , \quad (2.21)$$

where the three point function is calculated at the Gaussian fixed point. After substituting the leading order OPE expansion,

$$\mathcal{O}_i(x_1)\mathcal{O}_k(x_3) = \sum_{j=1}^{3}\frac{C_{ik}^j}{|x_{13}|^{\Delta-2\Delta_j}}\mathcal{O}_j(x_1) + \dots , \quad (2.22)$$

we get

$$\langle \mathcal{O}_i(x_1)\mathcal{O}_j(x_2)\rangle = \delta_{ij}\frac{N_i}{|x_{12}|^{2\Delta_i}} - \sum_{k=1}^{3}\frac{g_k\mu^{\epsilon_k}}{N}\Big(\frac{C_{ik}^j N_j}{|x_{12}|^{2\Delta_j}}\int\frac{d^dx_3}{|x_3|^{\Delta-2\Delta_j}} + (i\leftrightarrow j)\Big) + \mathcal{O}(g_k^2) .$$

$$(2.23)$$

Integrating out within a shell $\mu^{-1} < |x_3| < \mu_{\text{IR}}^{-1}$ between the subtraction scale $\mu$ and the IR cutoff $\mu_{IR}$ results in the following change

$$\delta\langle \mathcal{O}_i(x_1)\mathcal{O}_j(x_2)\rangle = -\frac{2\pi^{\frac{d}{2}}}{\Gamma\left(\frac{d}{2}\right)}\sum_{k=1}^{3}\frac{g_k\mu^{\epsilon_k}}{N}\Big(\frac{C_{ik}^j N_j}{|x_{12}|^{2\Delta_j}}\frac{\mu_{\text{IR}}^{-\epsilon_{ikj}} - \mu^{-\epsilon_{ikj}}}{\epsilon_{ikj}} + (i\leftrightarrow j)\Big) + \mathcal{O}(g_k^2) , \quad (2.24)$$

where $\epsilon_{ikj} = \epsilon_i + \epsilon_k - \epsilon_j$. Consider the case of equal epsilons, $\epsilon_i = \epsilon$. Then

$$\mu\frac{\partial}{\partial\mu}\langle \mathcal{O}_i(x_1)\mathcal{O}_j(x_2)\rangle = -\frac{2\epsilon}{N|x_{12}|^{2(d-\epsilon)}}\sum_{k=1}^{3}\tilde{g}_k\big(C_{ik}^j N_j + C_{jk}^i N_i\big) + \mathcal{O}(\epsilon^2) . \quad (2.25)$$

By $\mathcal{O}_i'$ we will denote primary operators in the weakly interacting theory at the fixed point. The Callan-Symanzyk equation for $\mathcal{O}_i'$ with the anomalous dimension $\gamma_i$ is given by

$$\mu\frac{\partial}{\partial\mu}\langle \mathcal{O}_i'(x_1)\mathcal{O}_i'(x_2)\rangle = -2\gamma_i\langle \mathcal{O}_i'(x_1)\mathcal{O}_i'(x_2)\rangle , \quad (2.26)$$

where to zeroth order in $\epsilon$

$$\mathcal{O}_i' = \sum_{k=1}^{3}V_i^k\mathcal{O}_k + \mathcal{O}(\epsilon) , \quad \langle \mathcal{O}_i'(x_1)\mathcal{O}_j'(x_2)\rangle = \frac{\sum_{k=1}^{3}V_i^k V_j^k N_k}{|x_{12}|^{2(d-\epsilon)}} = \frac{\delta_{ij}}{|x_{12}|^{2(d-\epsilon)}} . \quad (2.27)$$

The transition matrix $V_i^k$ is determined by requiring compatibility of (2.26) with (2.27), *i.e.*, using (2.27) and definition of $\mathcal{O}_i'$, we obtain

$$\mu\frac{\partial}{\partial\mu}\langle \mathcal{O}_m'(x_1)\mathcal{O}_n'(x_2)\rangle = -\frac{1}{|x_{12}|^{2(d-\epsilon)}}\sum_{i,j}\big(N_j V_n^j\gamma_{ji}V_m^i + N_i V_m^i\gamma_{ij}V_n^j\big) + \mathcal{O}(\epsilon^2) , (2.28)$$

$$\gamma_{ji} = \frac{2\epsilon}{N}\sum_k\tilde{g}_k C_{ik}^j = \frac{\partial\beta_j}{\partial g_i} + \epsilon\delta_{ij} .$$

Let us choose $V_m^i$ to be the eigenvectors of $\gamma_{ji}$ with eigenvalues $\gamma_m$, or equivalently, $\sum_{i=1}^{3} \gamma_{ji} V_m^i = \gamma_m V_m^j$, then[8]

$$\mu \frac{\partial}{\partial \mu} \langle \mathcal{O}_m'(x_1) \mathcal{O}_n'(x_2) \rangle = -\frac{2\gamma_m \delta_{mn}}{|x_{12}|^{2(d-\epsilon)}} + \mathcal{O}(\epsilon^2) . \tag{2.29}$$

Comparing to (2.26), we conclude that the anomalous dimensions are given by the eigenvalues of $\gamma_{ij}$.

The derivation of anomalous dimensions for non-equal $\epsilon_i$'s is similar. This is a nearly degenerate case, and therefore it is convenient to introduce an intermediate $\epsilon$ defined by $\epsilon_i = \epsilon + \delta\epsilon_i$ with $\delta\epsilon_i \sim \epsilon$. Next we rescale the operators $\mathcal{O}_i \to \mu^{\delta\epsilon_i} \mathcal{O}_i$. In particular, (2.25) for *rescaled* fields takes the form

$$\mu \frac{\partial}{\partial \mu} \langle \mathcal{O}_i(x_1) \mathcal{O}_j(x_2) \rangle = \frac{-2}{|x_{12}|^{2(d-\epsilon)}} \Big( -\delta\epsilon_j N_j \delta_{ij} + \frac{1}{N} \sum_{k=1}^{3} \epsilon_k \tilde{g}_k \big( C_{ik}^j N_j + C_{jk}^i N_i \big) \Big) + \mathcal{O}(\epsilon^2) . \tag{2.30}$$

Repeating the same steps as before, we conclude that the anomalous dimensions of $\mathcal{O}_m'$ in the nearly degenerate case are given by the eigenvalues, $\gamma_m$, of the matrix

$$\gamma_{ij} = -\delta\epsilon_i \delta_{ij} + \frac{2}{N} \sum_k \epsilon_k \tilde{g}_k C_{jk}^i = \frac{\partial \beta_i}{\partial g_j} + \epsilon \delta_{ij} . \tag{2.31}$$

To recapitulate, we find that non-locality of the model does not affect the expression for the anomalous dimensions. At leading order of the conformal perturbation theory they are given by the eigenvalues of the derivatives matrix $\frac{\partial \beta_i}{\partial g_j}$ evaluated at the fixed point. For the scaling dimensions of the operators $\mathcal{O}_i'$ we find

$$\Delta_m' = d - \epsilon + \gamma_m = d + \omega_m , \tag{2.32}$$

where $\omega_m$ are the eigenvalues of the derivatives matrix $\frac{\partial \beta_i}{\partial g_j}$.

The anomalous dimensions simplify in the case when all epsilons are equal. Then one of the three eigenvalues of $\gamma_{ij}$ can be readily derived using $\beta_i = 0$ and (2.9),

$$\sum_{j=1}^{3} \gamma_{ij} \tilde{g}_j = 2\epsilon \tilde{g}_i . \tag{2.33}$$

This eigenvalue corresponds to an irrelevant operator $\mathcal{O}' = \sum_{i=1}^{3} \tilde{g}_i \mathcal{O}_i$ with scaling dimension

$$\Delta' = d + \epsilon . \tag{2.34}$$

The scaling dimensions of two additional operators, $\mathcal{O}_\pm'$, are given by

$$\Delta_\pm' = d - \epsilon + \gamma_\pm , \tag{2.35}$$

$$\gamma_\pm = \frac{4\epsilon}{N} \left( \kappa_2 \pm \sqrt{\kappa_2^2 - \kappa_1} \right) , \quad \kappa_1 = 48(N+16)\tilde{g}_1\tilde{g}_2 , \quad \kappa_2 = \tilde{g}_1(m+14) + \tilde{g}_2(N-m+14) .$$

---

[8]Two terms within parenthesis in (2.28) are equal, because it follows from (2.7) that $N_j \gamma_{ji} = \frac{2\epsilon}{N} \sum_k \tilde{g}_k C_{ij}^k N_k$ is a symmetric matrix. Hence, for our choice of $V_m^i$ we get $\gamma_m \sum_j N_j V_n^j V_m^j = \gamma_n \sum_i N_i V_n^i V_m^i$, and therefore $\sum_j N_j V_n^j V_m^j = \delta_{mn}$, because degeneracy is lifted, *i.e.*, $\gamma_m \neq \gamma_n$ for $m \neq n$.

Note that they are real, because the radicand in the above expression is non negative,

$$\kappa_2^2 - \kappa_1 = g_1^2(m+14)^2 + g_2^2(N-m+14)^2 - 2g_1g_2\big[(m+14)(N-m+14) - 2(m+2)(N-m+2)\big] , \tag{2.36}$$

and therefore,[9]

$$\kappa_2^2 - \kappa_1 \geq \big[g_1(m+14) - g_2(N-m+14)\big]^2 \geq 0 . \tag{2.37}$$

For finite $N$ and $m$ the operators with scaling dimensions $\Delta'_+$ and $\Delta'_-$ are weakly irrelevant and relevant respectively. Thus the critical surface (subspace of irrelevant deformations at the interacting fixed point) has codimension 1 in the space of nearly marginal couplings $(g_1, g_2, g_3)$.

Using the last equation in (2.10), we deduce that in the large rank limit with $m/N$ fixed $\Delta_+ \to d$, $\Delta_- \to d - \epsilon$. The $\mathcal{O}'_+$ operator corresponds to an exactly marginal deformation associated with a line of fixed points (2.14) that emerges in the infinite $N$ limit. Each point on the conformal manifold has one weakly relevant and one weakly irrelevant deformation with scaling dimensions $d \mp \epsilon$ respectively.

Next consider the following single trace operators

$$\widetilde{\mathcal{O}}_1 = \phi_1^2 , \widetilde{\mathcal{O}}_2 = \phi_2^2 . \tag{2.38}$$

We remind the reader that by $\mathcal{O}_i$ we denote quartic operators, invariant under $O(m) \times O(N-m)$, while $\tilde{\mathcal{O}}_i$ above are quadratic in fields. Their scaling dimensions at the Gaussian fixed point are given by $\widetilde{\Delta}_1 = 2\Delta_{\phi_1}$, $\widetilde{\Delta}_2 = 2\Delta_{\phi_2}$. The two- and three-point functions satisfy

$$\langle \widetilde{\mathcal{O}}_j(x)\widetilde{\mathcal{O}}_k(0) \rangle = \delta_{jk} \frac{\widetilde{N}_k}{|x|^{2\widetilde{\Delta}_k}} ,$$
$$\widetilde{N}_1 = 2m , \quad \widetilde{N}_2 = 2(N-m) , \tag{2.39}$$

and

$$\langle \mathcal{O}_i(x_1)\widetilde{\mathcal{O}}_j(x_2)\widetilde{\mathcal{O}}_k(x_3) \rangle = \frac{\widetilde{C}_{ij}^k \widetilde{N}_k}{|x_{12}|^{\widetilde{\Delta}-2\widetilde{\Delta}_k}|x_{23}|^{\widetilde{\Delta}-2\Delta_i}|x_{13}|^{\widetilde{\Delta}-2\widetilde{\Delta}_j}},$$
$$\widetilde{\Delta} = \Delta_i + \widetilde{\Delta}_j + \widetilde{\Delta}_k, \tag{2.40}$$

where the non-zero OPE coefficients $\widetilde{C}_{ij}^k$ can be expressed in terms of $C_{ij}^k$

$$\widetilde{C}_{11}^1 = C_{13}^3 , \ \widetilde{C}_{31}^2 = C_{33}^2 , \ \widetilde{C}_{22}^2 = C_{32}^3 , \ \widetilde{C}_{32}^1 = C_{33}^1 . \tag{2.41}$$

As before they are related by

$$\widetilde{C}_{ij}^k = \widetilde{C}_{ik}^j \widetilde{N}_j / \widetilde{N}_k . \tag{2.42}$$

To calculate the leading order correction to the scaling dimensions of $\widetilde{\mathcal{O}}_i$ at the weakly interacting fixed point, we resort to the linear order conformal perturbation theory

$$\langle \widetilde{\mathcal{O}}_j(x_2)\widetilde{\mathcal{O}}_k(x_3) \rangle = \delta_{jk} \frac{N_k}{|x_{23}|^{2\widetilde{\Delta}_k}} - \sum_{i=1}^{3} \frac{g_i \mu^{\epsilon_i}}{N} \int_{\mu^{-1}} d^d x_1 \langle \mathcal{O}_i(x_1)\widetilde{\mathcal{O}}_j(x_2)\widetilde{\mathcal{O}}_k(x_3) \rangle + \mathcal{O}(g_k^2) , \tag{2.43}$$

---

[9]Stability of the fixed point requires $g_1$ and $g_2$ to be non-negative.

where the three point function is calculated at the Gaussian fixed point, and $\mu$ is a floating cutoff scale. In particular, using the leading order OPE expansion,

$$\mathcal{O}_i(x_1)\widetilde{\mathcal{O}}_j(x_2) = \sum_{k=1}^{2} \frac{\widetilde{C}_{ij}^k}{|x_{12}|^{\widetilde{\Delta}-2\widetilde{\Delta}_k}} \widetilde{\mathcal{O}}_k(x_2) + \dots \ , \tag{2.44}$$

one can calculate a small change in the two point function under variations in the floating cutoff scale $\mu$,

$$\delta\langle\widetilde{\mathcal{O}}_j(x_2)\widetilde{\mathcal{O}}_k(x_3)\rangle = -\sum_{i=1}^{3} \frac{g_i\mu^{\epsilon_i}}{N}\Big(\frac{\widetilde{C}_{ij}^k\widetilde{N}_k}{|x_{23}|^{2\widetilde{\Delta}_k}} \int_{\mu^{-1}}^{\mu_{\rm IR}^{-1}} \frac{d^dx_1}{|x_1|^{\widetilde{\Delta}-2\widetilde{\Delta}_k}} + (k\leftrightarrow j)\Big) + \mathcal{O}(g_k^2) \ . \tag{2.45}$$

In the case of equal epsilons, $\epsilon_i = \epsilon$, we obtain

$$\mu\frac{\partial}{\partial\mu}\langle\widetilde{\mathcal{O}}_j(x_2)\widetilde{\mathcal{O}}_k(x_3)\rangle = -\frac{2\epsilon}{N\,|x_{23}|^{d-\epsilon}}\sum_{i=1}^{3}\tilde{g}_i\big(\widetilde{C}_{ij}^k\widetilde{N}_k + \widetilde{C}_{ik}^j\widetilde{N}_j\big) + \mathcal{O}(\epsilon^2) \ . \tag{2.46}$$

At this point we introduce primary operators in the weakly interacting CFT, i.e., $\widetilde{\mathcal{O}}_i' = \sum_{k=1}^{2}\widetilde{V}_i^k\widetilde{\mathcal{O}}_k + \mathcal{O}(\epsilon)$ with the anomalous dimension $\widetilde{\gamma}_i$. The Callan-Symanzyk equation for $\widetilde{\mathcal{O}}_i'$ is given by

$$\mu\frac{\partial}{\partial\mu}\langle\widetilde{\mathcal{O}}_i'(x_1)\widetilde{\mathcal{O}}_i'(x_2)\rangle = -2\widetilde{\gamma}_i\,\langle\widetilde{\mathcal{O}}_i'(x_1)\widetilde{\mathcal{O}}_i'(x_2)\rangle \ , \tag{2.47}$$

where

$$\langle\widetilde{\mathcal{O}}_i'(x_1)\widetilde{\mathcal{O}}_j'(x_2)\rangle = \frac{\sum_{k=1}^{2}\widetilde{V}_i^k\widetilde{V}_j^k\widetilde{N}_k}{|x_{12}|^{d-\epsilon}} = \frac{\delta_{ij}}{|x_{12}|^{d-\epsilon}} \ . \tag{2.48}$$

The transition matrix $\widetilde{V}_i^k$ is determined by requiring compatibility of (2.47) with (2.48),

$$\mu\frac{\partial}{\partial\mu}\langle\widetilde{\mathcal{O}}_m'(x_1)\widetilde{\mathcal{O}}_n'(x_2)\rangle = -\frac{1}{|x_{12}|^{d-\epsilon}}\sum_{k,j=1}^{2}\Big(\widetilde{N}_k\widetilde{V}_n^k\widetilde{\gamma}_{kj}\widetilde{V}_m^j + \widetilde{N}_j\widetilde{V}_m^j\widetilde{\gamma}_{jk}\widetilde{V}_n^k\Big) + \mathcal{O}(\epsilon^2) \tag{2.49}$$

$$\widetilde{\gamma}_{kj} = \frac{2\epsilon}{N}\sum_{i=1}^{3}\tilde{g}_i\widetilde{C}_{ij}^k \ .$$

Let us choose $\widetilde{V}_m^j$ to be the eigenvectors of $\widetilde{\gamma}_{kj}$ with eigenvalues $\widetilde{\gamma}_m$, or equivalently, $\sum_{j=1}^{2}\widetilde{\gamma}_{kj}\widetilde{V}_m^j = \widetilde{\gamma}_m\widetilde{V}_m^k$, then[10]

$$\mu\frac{\partial}{\partial\mu}\langle\widetilde{\mathcal{O}}_m'(x_1)\widetilde{\mathcal{O}}_n'(x_2)\rangle = -\frac{2\widetilde{\gamma}_m\delta_{mn}}{|x_{12}|^{d-\epsilon}} + \mathcal{O}(\epsilon^2). \tag{2.50}$$

Comparing to (2.26), we conclude that the anomalous dimensions are given by the eigenvalues of $\widetilde{\gamma}_{kj}$,

$$\widetilde{\gamma}_\pm = \frac{\epsilon}{N}\left(C_{13}^3\tilde{g}_1 + C_{32}^3\tilde{g}_2 \pm \sqrt{\left(C_{13}^3\tilde{g}_1 - C_{32}^3\tilde{g}_2\right)^2 + 4C_{33}^1C_{33}^2\tilde{g}_3^2}\right) \ ,$$

$$\widetilde{V}_\pm = \left(\frac{C_{13}^3\tilde{g}_1 - C_{32}^3\tilde{g}_2 \pm \sqrt{\left(C_{13}^3\tilde{g}_1 - C_{32}^3\tilde{g}_2\right)^2 + 4C_{33}^1C_{33}^2\tilde{g}_3^2}}{2C_{33}^2\tilde{g}_3} \ , \ 1\right) \ . \tag{2.51}$$

---

[10]Two terms within parenthesis in (2.49) are equal, because it follows from (2.42) that $\widetilde{N}_j\widetilde{\gamma}_{ji}$ is a symmetric matrix. Hence, for our choice of $\widetilde{V}_m^i$ we get $\widetilde{\gamma}_m\sum_k\widetilde{N}_k\widetilde{V}_n^k\widetilde{V}_m^k = \widetilde{\gamma}_n\sum_j\widetilde{N}_j\widetilde{V}_n^j\widetilde{V}_m^j$, and therefore $\sum_k\widetilde{N}_k\widetilde{V}_n^k\widetilde{V}_m^k = \delta_{mn}$, because degeneracy is lifted, i.e., $\widetilde{\gamma}_m \neq \widetilde{\gamma}_n$ for $m \neq n$.

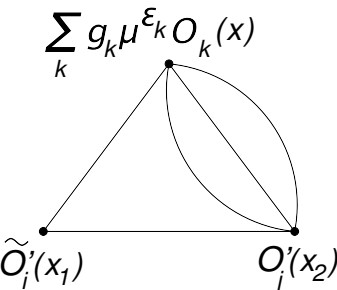

**Figure 2**. Linear order contribution to the correlator $\langle \widetilde{\mathcal{O}'}_i(x_1)\mathcal{O}'_j(x_2)\rangle$. Solid lines represent propagators of the scalar fields. Integration is done over the insertion point $x$.

Note that the anomalous dimensions, $\widetilde{\gamma}_\pm$, are manifestly real for all $m$ and $N$.

For non-equal $\epsilon_i$'s, the scaling dimensions of two operators $\widetilde{\mathcal{O}}'_i$ can be similarly derived

$$\widetilde{\Delta}'_m = \frac{d}{2} + \widetilde{\omega}_m \; , \tag{2.52}$$

where $\widetilde{\omega}_m$ are the eigenvalues of the $2\times 2$ matrix

$$\widetilde{\gamma}_{kj} = -\frac{\epsilon_k}{2}\delta_{kj} + \frac{2}{N}\sum_{i=1}^3 \epsilon_i \tilde{g}_i \widetilde{C}^k_{ij} \; . \tag{2.53}$$

In the vicinity of $\alpha = 1$, i.e. for $\epsilon_1 \approx \epsilon_2$ the eigenvalues of (2.53) are perturbatively close to (2.51), which are real. Therefore eigenvalues of (2.53) will also be real, at least until two of them collide.

To conclude, we have calculated anomalous dimensions of all quadratic and quartic operators at the interacting fixed points, at leading order in the conformal perturbation theory. Assuming the fixed point is stable, all scaling dimensions are real, as it is necessary for unitarity of the IR theory.

## 2.4 Tests of conformal invariance

The fixed point QFT is certainly scale invariant, but it is not necessarily a CFT. In this section we perform a number of tests to provide evidence that the scale symmetry in our models is enhanced to the full conformal group. For simplicity we consider the case of equal epsilons only, $\epsilon_i = \epsilon$. The case of non equal epsilons is similar.

We start by calculating the two point function of primaries at the fixed point, $\langle \widetilde{\mathcal{O}'}_i\mathcal{O}'_j\rangle$. These correlators could be non-zero if the model is scale invariant but non-conformal. However, it must vanish if the theory exhibits full conformal symmetry.

To linear order in $\epsilon$ to evaluate $\langle \widetilde{\mathcal{O}'}_i\mathcal{O}'_j\rangle$ we should calculate the diagram shown in Fig. 2. Up to an overall factor, we have

$$\text{Fig.2} \propto \frac{\epsilon\,\mu^\epsilon}{|x_{12}|^{\frac{d-\epsilon}{2}}}\int d^dx \frac{1}{|x-x_1|^{\frac{d-\epsilon}{2}}|x-x_2|^{\frac{3(d-\epsilon)}{2}}} = \pi^{\frac{d}{2}}\frac{\epsilon\,(|x_{12}|\mu)^\epsilon}{|x_{12}|^{\frac{3(d-\epsilon)}{2}}}\frac{\Gamma\!\left(\frac{d-2\epsilon}{2}\right)\Gamma\!\left(\frac{d+\epsilon}{4}\right)\Gamma\!\left(\frac{3\epsilon-d}{4}\right)}{\Gamma\!\left(\frac{d-\epsilon}{4}\right)\Gamma\!\left(\frac{3(d-\epsilon)}{4}\right)\Gamma(\epsilon)} \; , \tag{2.54}$$

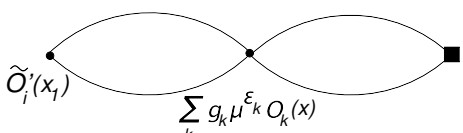

**Figure 3**. Linear order contribution to the correlation function of $\widetilde{\mathcal{O}}'_i$ with a primary built off two scalar fields and any number of derivatives (solid square). Solid lines represent propagators of the scalar fields. Integration is done over the insertion point $x$.

where the proportionality constant is some function of $\tilde{g}_k$, $N$ and $m$, and we used the following identity

$$\int d^d x \frac{1}{|x-x_1|^\alpha |x-x_2|^\beta} = \pi^{d/2} \frac{\Gamma\left(\frac{\alpha+\beta-d}{2}\right)}{\Gamma(\alpha/2)\Gamma(\beta/2)} \frac{\Gamma\left(\frac{d-\alpha}{2}\right)\Gamma\left(\frac{d-\beta}{2}\right)}{\Gamma(d-\alpha/2-\beta/2)} |x_{12}|^{d-\alpha-\beta} \ . \quad (2.55)$$

Gamma function $\Gamma(\epsilon)$ in the denominator of (2.54) introduces one extra power of $\epsilon$ and therefore to linear order in $\epsilon$ that expression vanishes, yielding

$$\langle \widetilde{\mathcal{O}}'_i(x_1)\mathcal{O}'_j(x_2) \rangle = 0 + \mathcal{O}(\epsilon^2) \ . \quad (2.56)$$

As can be seen from (2.55), this result also holds if $\mathcal{O}'_j(x_2)$ is replaced with a scalar operator which is quartic in fields and has any number of derivatives. Likewise the linear order correction to the correlation function of $\widetilde{\mathcal{O}}'_i$ with any operator which is more than quartic in fields also vanishes identically.

Consider now the correlation function of $\widetilde{\mathcal{O}}'_i$ with a primary operator, which is quadratic in fields and which includes any number of derivatives. This correlator vanishes to zeroth order in the coupling constant, because the Gaussian theory is conformal. The linear order correction is shown in Fig. 3, it factorizes into a product of two sub-diagrams. One of them vanishes, because it is proportional to the correlation function of two distinct primaries in the Gaussian model.

The upshot of this calculation is that up to linear order in $\epsilon$ the two point function of $\widetilde{\mathcal{O}}'_i$ with various operators is compatible with the conformal symmetry – all two-point correlators of operators with non-equal dimensions vanish. This result can be used to show that the three point function,

$$\langle \widetilde{\mathcal{O}}'_i(x_1)\widetilde{\mathcal{O}}'_j(x_2)\widetilde{\mathcal{O}}'_k(x_3) \rangle = \sum_{\ell_1,\ell_2,\ell_3=1}^{2} \widetilde{V}_i^{\ell_1} \widetilde{V}_j^{\ell_2} \widetilde{V}_k^{\ell_3} \langle \widetilde{\mathcal{O}}_{\ell_1}(x_1)\widetilde{\mathcal{O}}_{\ell_2}(x_2)\widetilde{\mathcal{O}}_{\ell_3}(x_3) \rangle \ . \quad (2.57)$$

is conformal up to linear order in $\epsilon$.

Indeed, integrating out distances within the shell $\mu^{-1} < |x| < \mu_{\text{IR}}^{-1}$ and using (2.44) results in the following change

$$\delta\langle \widetilde{\mathcal{O}}_{\ell_1}(x_1)\widetilde{\mathcal{O}}_{\ell_2}(x_2)\widetilde{\mathcal{O}}_{\ell_3}(x_3) \rangle = \quad (2.58)$$
$$-\sum_{k=1}^{3} \frac{g_k \mu^\epsilon}{N} \Big( \sum_{m=1}^{2} \tilde{C}_{k\ell_3}^m \langle \widetilde{\mathcal{O}}_{\ell_1}(x_1)\widetilde{\mathcal{O}}_{\ell_2}(x_2)\widetilde{\mathcal{O}}_m(x_3) \rangle_0 \int_{\mu^{-1}}^{\mu_{\text{IR}}^{-1}} \frac{d^d x}{|x|^{d-\epsilon}} + (1 \leftrightarrow 3, 2 \leftrightarrow 3) \Big) + \mathcal{O}(g_k^2) \ .$$

Or equivalently,

$$\mu\frac{\partial}{\partial\mu}\langle\widetilde{\mathcal{O}}_{\ell_1}(x_1)\widetilde{\mathcal{O}}_{\ell_2}(x_2)\widetilde{\mathcal{O}}_{\ell_3}(x_3)\rangle = -\sum_{m=1}^{2}\tilde{\gamma}_{m\ell_3}\langle\widetilde{\mathcal{O}}_{\ell_1}(x_1)\widetilde{\mathcal{O}}_{\ell_2}(x_2)\widetilde{\mathcal{O}}_m(x_3)\rangle_0$$
$$+ (1 \leftrightarrow 3,\, 2 \leftrightarrow 3) + \mathcal{O}(g_k^2) , \qquad (2.59)$$

where $\tilde{\gamma}_{m\ell_3}$ is defined in (2.49). Plugging it back into (2.57) and using the fact that $V_i^\ell$ are the eigenvectors of $\tilde{\gamma}_{m\ell}$ with eigenvalues $\tilde{\gamma}_i$, we conclude that the three point function (2.57) takes the following general form

$$\langle\widetilde{\mathcal{O}'}_i(x_1)\widetilde{\mathcal{O}'}_j(x_2)\widetilde{\mathcal{O}'}_k(x_3)\rangle \sim \sum \frac{\mu^{-\tilde{\gamma}_i-\tilde{\gamma}_j-\tilde{\gamma}_k}}{|x_{12}|^{\frac{d-\epsilon}{2}+\alpha_{ijk}}|x_{13}|^{\frac{d-\epsilon}{2}+\alpha_{ikj}}|x_{23}|^{\frac{d-\epsilon}{2}+\alpha_{jki}}} , \qquad (2.60)$$

where the sum includes all possible $\alpha$'s which satisfy $\alpha_{ijk} = \alpha_{jik} \sim \epsilon$ and $\alpha_{ijk}+\alpha_{ikj}+\alpha_{jki} = \tilde{\gamma}_i + \tilde{\gamma}_j + \tilde{\gamma}_k$. In particular, for $i = j = k$, we get only one possible term with $\alpha_{iii} = \gamma_i$, and the associated three point function at linear order in $\epsilon$ is necessarily conformal. We should only consider the case when one of the three indices $i, j, k$ is different from the other two, e.g., $i = j = +$ and $k = -$. In the limit $x_1 \to x_2$ the leading order singularity takes the form

$$\langle\widetilde{\mathcal{O}'}_+(x_1)\widetilde{\mathcal{O}'}_+(x_2)\widetilde{\mathcal{O}'}_-(x_3)\rangle \xrightarrow[x_1\sim x_2]{} \frac{\mu^{-\tilde{\gamma}_i-\tilde{\gamma}_j-\tilde{\gamma}_k}}{|x_{12}|^{\frac{d-\epsilon}{2}+\alpha_{++-}}|x_{13}|^{d-\epsilon+2\alpha_{+-+}}} . \qquad (2.61)$$

Since $x_1 \sim x_2$, one can substitute an appropriate OPE for the $\widetilde{\mathcal{O}'}_+(x_1)\widetilde{\mathcal{O}'}_+(x_2)$ on the left hand side. However, we have shown that the two-point function of $\widetilde{\mathcal{O}'}_-$ with various operators respects conformal symmetry to linear order in $\epsilon$. Thus, only the term proportional to $\widetilde{\mathcal{O}'}_-$ in the OPE contributes in this limit. As a result, the left hand side scales as $1/|x_{13}|^{d-\epsilon+2\gamma_-}$. In particular, $\alpha_{+-+} = \gamma_-$, $\alpha_{++-} = 2\gamma_+ - \gamma_-$ and the three point function $\langle\widetilde{\mathcal{O}'}_i\widetilde{\mathcal{O}'}_j\widetilde{\mathcal{O}'}_k\rangle$ is necessarily conformal up to linear order in $\epsilon$.

# 3 Thermal physics

To understand the unbroken symmetries of the critical model at finite temperature we consider the effective potential, $V_{\text{eff}}$. To leading order in $\epsilon_i$, thermal fluctuations simply induce quadratic terms in addition to the quartic potential (2.8). Starting from the thermal correlation function for the generalized free fields we find

$$\langle\phi_j^a(\tau,\vec{x})\phi_i^c(0)\rangle_\beta = \sum_{m=-\infty}^{\infty}\frac{\delta^{ac}\delta_{ij}}{[(\tau+m\beta)^2+\vec{x}^2]^{\Delta_{\phi_i}}} \quad \Rightarrow \quad \langle\phi_i^2\rangle_\beta = Nx_i\frac{2\,\zeta(2\Delta_{\phi_i})}{\beta^{2\Delta_{\phi_i}}} . \qquad (3.1)$$

Together with the interaction terms in (2.8) this leads to the following effective potential for the zero mode

$$V_{\text{eff}}(\phi_1,\phi_2;\beta) = \mathcal{M}_{\phi_1}(\beta)\phi_1^2 + \mathcal{M}_{\phi_2}(\beta)\phi_2^2 + \frac{g_1\mu^{\epsilon_1}}{N}(\phi_1^2)^2 + \frac{g_2\mu^{\epsilon_2}}{N}(\phi_2^2)^2 + \frac{g_3\mu^{\epsilon_3}}{N}\phi_1^2\phi_2^2 , \qquad (3.2)$$

where we dropped terms suppressed by the higher powers of $\epsilon$ , and

$$\mathcal{M}_{\phi_1}(\beta) = 2\frac{g_1\mu^{\epsilon_1}}{N}\Big(1+\frac{2}{Nx_1}\Big)\langle\phi_1^2\rangle_\beta + \frac{g_3\mu^{\epsilon_3}}{N}\langle\phi_2^2\rangle_\beta , \qquad (3.3)$$

$$\mathcal{M}_{\phi_2}(\beta) = 2\frac{g_2\mu^{\epsilon_2}}{N}\Big(1+\frac{2}{Nx_2}\Big)\langle\phi_2^2\rangle_\beta + \frac{g_3\mu^{\epsilon_3}}{N}\langle\phi_1^2\rangle_\beta. \qquad (3.4)$$

In the absence of quadratic terms[11], $\mathcal{M}_{\phi_1} = \mathcal{M}_{\phi_2} = 0$, the potential reaches its minimum value at $\phi_i^2 = 0$. Hence, the system exhibits full $O(m) \times \mathcal{O}(N - m)$ symmetry at zero temperature. However, finite temperature effects may break the symmetry provided that $\mathcal{M}_{\phi_i} < 0$. If that occurs, the higher order perturbative corrections cannot restore the symmetry, because multiloop quadratic terms are suppressed by additional powers of $\epsilon_i$, whereas terms with higher powers of fields will be subdominant in the vicinity of the origin $\phi_i^2 = 0$. Therefore, to prove that the symmetry is broken at finite temperature, it is enough to show that the model admits a fixed point where one of the quadratic terms in the effective potential (3.2) becomes negative.

Thermal expectation values $\langle \phi_i^2 \rangle_\beta$, given by (3.1), are positive, and therefore $\mathcal{M}_{\phi_i}$ can only become negative if some of the critical couplings $g_i$ are negative. Since the couplings $g_1$ and $g_2$ must be positive to ensure stability of the model, the only scenario would be $g_3 < 0$, while $4g_1 g_2 \geq g_3^2$ to exclude the runaway behavior. In fact, the potential is always bounded from below as long as the fixed point equations (2.10) are satisfied [12, 50]. We verified explicitly (numerically or analytically) that the stability condition $4g_1 g_2 \geq g_3^2$ holds in all the examples discussed below.

We first analyze the $N \to \infty$ limit with $x_1 = m/N$ kept fixed. The effective potential in this case assumes the following form[12]

$$V_{\text{eff}}(z; \beta) = 2\frac{\mu^\epsilon}{N}\left(\langle z \rangle_\beta z + \frac{z^2}{2}\right), \quad z = \sqrt{g_1}\phi_1^2 \pm \sqrt{g_2}\phi_2^2 . \tag{3.5}$$

Here we used the leading order relation between the critical couplings, $g_3 = \pm 2\sqrt{g_1 g_2}$, which follows from (2.14). For positive $g_3$, $\langle z \rangle_\beta > 0$, and the effective potential is minimized by $\phi_i^2 = 0$ because $\phi_i^2$ cannot be negative. For negative $g_3$ the situation is more nuanced. Provided constraints $\phi_i^2 > 0$ are satisfied, the minimum is reached at

$$z = \sqrt{g_1}\phi_1^2 - \sqrt{g_2}\phi_2^2 = -\langle z \rangle_\beta = (x_2\sqrt{g_2} - x_1\sqrt{g_1})\frac{2\zeta(d/2)}{\beta^{d/2}} , \quad \text{for} \quad g_3 < 0 , \tag{3.6}$$

where in the expression for $\langle z \rangle_\beta$ we dropped $\epsilon_i$-suppressed terms. This defines a one-dimensional family of minima in the space of fields – a straight line in the coordinates $\phi_1^2$ and $\phi_2^2$. In the special case when $x_1\sqrt{g_1} = x_2\sqrt{g_2}$, the parameter $\langle z \rangle_\beta = 0$, and there is trivial solution $\phi_1^2 = \phi_2^2 = 0$, together with the non-trivial ones $\phi_1^2 = \phi_2^2 > 0$. Hence in this case we can not establish symmetry breaking unless $1/N$ corrections are taken into account. But in all other cases (3.6), together with the constraints $\phi_i^2 > 0$, necessarily yields the solutions with at least one or both fields being non-zero.

When $N$ is finite, one in principle needs to minimize (3.2) with the additional conditions $\phi_i^2 > 0$. Provided one of the masses is negative, say $\mathcal{M}_{\phi_1} < 0$, the minimum is given by

$$\begin{pmatrix} \phi_1^2 \\ \phi_2^2 \end{pmatrix} = \frac{-N\mu^{-\epsilon_1}}{2g_1}\begin{pmatrix} \mathcal{M}_{\phi_1} \\ 0 \end{pmatrix} . \tag{3.7}$$

---

[11] Note that $\mathcal{M}_{\phi_i}$ has the non-canonical scaling dimension of $\frac{d+\epsilon_i}{2}$.

[12] As we explained in section 2.1, in the large rank limit the couplings must satisfy $\alpha \to 1$, and therefore $\epsilon_1 = \epsilon_2 = \epsilon$.

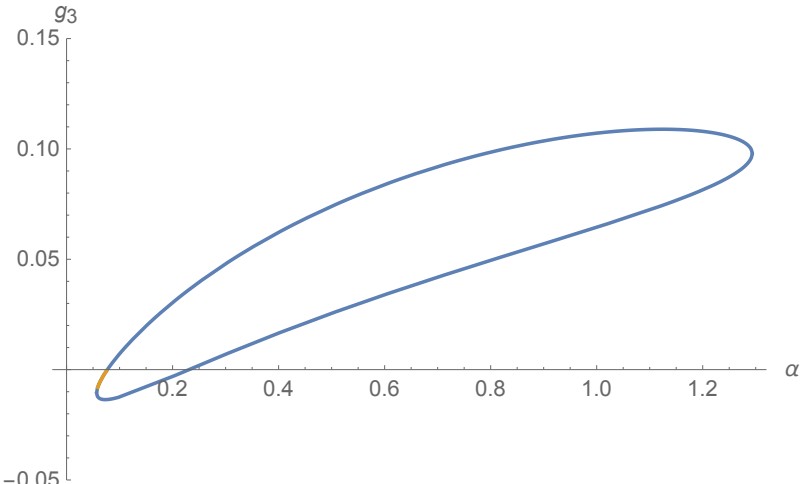

**Figure 4**. Critical value of $g_3$ as a function of $\alpha$ for $m = 1, N = 6$ case. Orange region corresponds to $\mathcal{M}_{\phi_1} < 0$.

while the only point with the unbroken symmetry $\phi_1^2 = \phi_2^2 = 0$ is not a minimum. It is thus sufficient to show there are critical points with negative $\mathcal{M}_{\phi_1}$, which we do numerically. Thus typically there are two solutions for critical $\tilde{g}_i$ for a given value of $\alpha$. There are points with $g_3 < 0$ for which $\mathcal{M}_{\phi_1}$ is positive, but there are also those where $\mathcal{M}_{\phi_1} < 0$. We illustrate that in Fig. 4 for $m = 1, N = 6$. The behavior for $m = 1$ and $N > 6$ as well for $m = 2, N \geq 7$, and more generally $N \geq \min(m + 5, 2m)$, is similar (we use the notations $m \leq N - m$). There are always values of $\alpha \leq \alpha_c$ such that $g_3, \mathcal{M}_{\phi_1} < 0$, and hence the corresponding critical points exhibit persistent symmetry breaking.

## 4 Discussion

In this paper we constructed and studied a three-dimensional model that exhibits global continuous symmetry breaking at arbitrarily large temperatures. To our knowledge this is the first example of a UV complete unitary 3d model exhibiting persistent breaking of a continuous global symmetry. It bypasses the Coleman-Hohenberg-Mermin-Wagner no-go theorem [29–31] by incorporating non-local interactions.[13]

Our model is comprised of two copies of the generalized free fields $\phi_{1,2}$ in the fundamental representation of $O(m)$ and $O(N - m)$ respectively. The bare dimensions are tuned such that quartic interactions are weakly relevant and consequently the model can be analyzed using conformal perturbation theory. An interacting theory is obtained by turning on all quartic interactions preserving global $O(m) \times O(N - m)$ symmetry. Besides $N$ and $m$ the model is parametrized by $\alpha = \epsilon_1/\epsilon_2$, where $\Delta_{1,2} = (d - \epsilon_{1,2})/4$ are bare scaling dimensions of fundamental fields and we work in the regime $\epsilon_i \ll 1$ to leading order in $\epsilon_i$.

---

[13]Placing a theory on a curved spacetime is another way to bypass the CHMW theorem. For instance, the O(N) model in AdS evades it [51], but at high temperatures the symmetry is restored in this model.

We found that the resulting IR flow terminates at the fixed points located in the perturbative vicinity of the origin. In this sense our model is similar to the Banks-Zaks construction [52]. In the infinite $N$ limit, we find a conformal manifold, which is in fact a circle. It includes points where the coupling constant $g_3$, that controls the coupling between $O(m)$ and $O(N-m)$ fields, vanishes, and the model degenerates into two decoupled long range models, one of which is free and the other critical. It also includes a point where global symmetry is enhanced to $O(N)$. For us of particular interest are the fixed points with $g_3 < 0$, as this is a necessary (but not sufficient) condition for the symmetry to be broken at finite temperature. For large but finite $N$ we have a continuous family of CFTs parametrized by $\epsilon_i$. The perturbative fixed points with $g_3 < 0$ also survive for finite $N$. This behavior continues up to small values of $N$. Assuming $m \leq N - m$, we find that $g_3 < 0$ fixed points exist for any $N \geq \max(m+5, 2m)$.

At zero temperature, $\langle \phi_i^2 \rangle = 0$ and the full $O(m) \times O(N-m)$ symmetry is preserved. We argue this is a unitary theory with full conformal symmetry, not merely a set of scale-invariant fixed points. For that purpose we show that the anomalous dimensions are real. We also study two- and three-point functions. Working at the linear order in $\epsilon_{1,2}$ we demonstrate that two-point functions of the operators with different scaling dimensions vanish, as required by full conformal symmetry. Furthermore, within the same approximation three-point functions exhibit the form consistent with the full conformal invariance. While our results do not constitute a proof, they strongly suggest the interacting $O(m) \times O(N-m)$ theory is unitary and conformal, which extends previous results [37] arguing for conformality of the interacting long range $O(m)$ model.

At finite temperature $T$ certain fixed points with $g_3 < 0$, which appear for all $N \geq \max(m+5, 2m)$ and particular $\alpha$, exhibit spontaneous symmetry breaking $O(m) \times O(N-m) \to O(m-1) \times O(N-m)$. Since the theory is scale-invariant, symmetry breaking persists at all temperatures. Thus our model provides a generalization of [8] which considered the case of $m = 1$ and reported persistent breaking of discrete $\mathbb{Z}_2 = O(1)$ symmetry. The crucial ingredient in our construction is the non-local interactions, which is necessary to circumvent the CHMW no-go theorem. In light of the previous attempts to achieve the persistent breaking behavior with local interactions, which were either UV-incomplete [1], or required fractional dimensions [11, 12] where unitarity is violated [34], or required strictly infinite $N$ [14, 16], one naturally wonders if the persistent breaking of a discrete symmetry is possible in a UV-complete, unitary, relativistic three-dimensional theory with local interactions? A candidate for such a model was suggested in [11, 12]. However, it is hard to establish the existence of PSB phenomenon in their model directly in $2 + 1$ dimensions. A similar question regarding continuous symmetry is answered by the CHMW theorem which prohibits such a behavior. Interpreting results of this paper as an evidence that non-locality of interactions is more important than the discreteness of symmetry to achieve persistent breaking, we tend to conclude that the answer to the question above is negative. It would therefore be an important step to generalize the CHMW no-go result to include discrete symmetries.

Our results prompt further research. Recently, generalized free fermionic models, perturbed by four-fermionic interaction, have been considered in [53]. It would be interesting

to see if such non-local models lead to conformal fixed points which can exhibit persistent symmetry breaking. Beyond the fermionic QFTs one can ask a similar question already for the lattice settings. It is well-known that in local, i.e. short range lattice models symmetries are always restored at sufficiently large temperatures, of the order of the inverse lattice size [54]. This is also the case for models with exponentially decaying interactions, which should be regarded as local in all physical senses. The question our work poses is to see if for lattices with the polynomial interactions the results of [54] break down and if persistent breaking is possible.

**Acknowledgements** We thank A. Avdoshkin, S. Chaudhuri, C. Choi, Z. Komargodski, E. Rabinovici for helpful discussions and correspondence. NC, RS and MS are grateful to the Israeli Science Foundation Center of Excellence (grant No. 2289/18) and the Quantum Universe I-CORE program of the Israel Planning and Budgeting Committee (grant No. 1937/12) for continuous support of our research. NC is grateful for the support from the Yuri Milner scholarship. AD is supported by the NSF under grant PHY-2013812. AD also acknowledges KITP for hospitality. The research at KITP was supported in part by the National Science Foundation under Grant No. PHY-1748958. The work of MG is partially supported by DOE grant DE-SC0011842.

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
