# Peer review of "A model of persistent breaking of continuous symmetry"

_SciPost Physics_

## Round 1 · Referee Report · Anonymous (Referee 1) · 2022-2-5

Report

The search for conformal field theories (CFTs) that display persistent breaking of global symmetries has emerged as an interesting research direction. In some recent work involving some of the authors of this manuscript, biconical models in $d=4-\varepsilon$ were considered with the surprising conclusion that they display persistent symmetry breaking. These models were defined in fractional dimensions, and although large-$N$ results valid in arbitrary $d$ were also obtained, the undisputed discovery of models with persistent symmetry breaking in physical (integer) dimensions, particularly $d=3$, was still missing.

This manuscript considers a long-range model with global symmetry $O(m)\times O(N-m)$ defined in arbitrary dimension $d$ that is shown to display persistent symmetry breaking. The authors declare this result as the first example of such a model in integer $d$ and emphasize the $d=3$ case. Although the authors clearly state their assumptions, e.g. the non-locality of the model, the construction feels like it succeeds by "moving the goalposts", in some sense.

Let me explain what I mean. To start, I have no doubt that the authors' calculations are correct. However, the existence of persistent symmetry breaking is a direct result of an extra parameter, called $\alpha$ by the authors, which is freely tunable as a result of their very consideration of long-range models. The fact that this parameter can be freely chosen is a weak point of the paper in my view. One may say that $\alpha$ should not be treated differently than $m, N$, which are also freely tunable parameters, but the point of my argument is that a parameter like $\alpha$, which is related to scaling dimensions of operators, should not be freely tunable but dynamical, i.e. determined by the dynamics of the theory. I would view that as a requirement, for otherwise the example is rather artificial.

This manuscript is well-written and deserves to be published in a suitable form. Before publication, I would recommend that the authors invert the order of some sentences to describe the context first and their claim second. Even the first sentence of the abstract is missing key context, which is provided (but not emphasized) in the second sentence. In situations like these, one should start by providing proper context before announcing a result and not the other way around as the authors have unfortunately chosen in key parts of their manuscript.

The changes I am suggesting are basically limited to the abstract and conclusion. I think that the introduction describes the situation adequately well. The authors should reconsider the first two sentences of their abstract and the first paragraph of their conclusion.

I would like the authors to also explain why they limit the spacetime dimensionality of their theory to $1<d<4$, when in fact their theory (2.1) can be defined in arbitrary $d$. After these small modifications, I would like to reconsider the manuscript for publication.
  • validity: high
  • significance: ok
  • originality: good
  • clarity: high
  • formatting: excellent
  • grammar: good

Author:  Noam Chai  on 2022-02-22  [id 2235]

(in reply to Report 1 on 2022-02-05)
Category:
remark
correction

We would like to thank our referee for carefully reading our manuscript, raising questions and providing valuable comments. In the new version of our manuscript we thoroughly addressed all the comments and suggestions raised by the referee:

  1. We accept the referee's comment to rewrite the first two sentences of the abstract and the first paragraph of the conclusion. As suggested, we emphasized the context first and the claims second.

  2. $\alpha$ is in fact an arbitrary parameter, just like $N$ or $m$. Since UV divergences are local, the wave function renormalization can only affect fields which exhibit local kinetic terms. This can also be seen within Wilsonian renormalization. Hence, the scaling dimension of the field with non-local kinetic term is protected. In particular, the fields in our case do not acquire anomalous dimensions, and $\alpha$ is unaffected by the dynamics of the model. To clarify this point, we explained in the second section how $\alpha$ is defined.

  3. The range of $d$ ensures the stability of the model, and is fixed by the unitarity bound ($d/4>\Delta_\phi \geq \frac{d-2}{2}$). We added a comment in the second section on how $d$ is determined.

---

## Round 1 · Referee Report · Anonymous (Referee 2) · 2022-3-19

Report

The paper is about the phenomenon of persistent symmetry breaking, i.e. the spontaneous breaking of a symmetry that does not get restored at any finite temperature.

The goal is demonstrating the existence of models that exhibit persistent breaking of a continuous symmetry in 2+1 d . To do so it is necessary to evade the CHMW theorem that would forbid breaking of a continuous symmetry at finite temperature in 2+1 d. Therefore the authors consider a model with non-local interactions.

The paper is interesting and worth publishing on SciPost. However there are some changes/clarifications needed prior to publications:

  • In the intro the phrasing "to our knowledge there were no prior known examples of the non-SUSY finite N models admitting an exactly marginal deformation" seems to imply that the model studied here provides such an example. However I saw no evidence of this in the paper. In the discussion in subsection 2.2 the authors are --correctly-- careful to distinguish between two parameter family of theories and conformal manifolds. They say that the family of CFTs persists at finite N. But of course to show that is a conformal manifold one needs to exhibit an exactly marginal operator. This is studied in the following subsection 2.3, but the existence of an exactly marginal operator is only showed "in the large rank limit with m/N fixed" (quoting from pag.12). So what is the evidence for the finite N conformal manifold? If there is no such evidence, the phrasing in the intro needs to change.

  • The authors feel like they need to repeat the derivation of well-known conformal perturbation theory (CPT) formulas (see e.g. the reviews in https://arxiv.org/abs/1603.04444 and the review and the references in https://arxiv.org/abs/1703.05325) because the theory is non local. But it is intrinsic in the setup of CPT that one deals with operators whose dimensions are taken as free parameters, that we choose close to marginality. So in a sense one is always dealing with a non-local theory. Also in the standard derivations clearly locality is never used. So I did not understand the need for this re-derivation, and also the lack of references to previous literature. I suggest they explain better why they think non-locality matters, or say explicitly that they are just reviewing known material and give appropriate references to the literature.

  • The last paragraph in pag. 19 is rather confusing. We read there that this paper provides the generalization of reference [8] from discrete to continuous symmetry. But then we read "one naturally wonders if the persistent breaking of a discrete symmetry is possible in a UV-complete, unitary, relativistic three-dimensional theory with local interactions?". From the first part of the paragraph I had understood that this is exactly what is done in [8]. When talking about [8] the authors should say which of the desiderata is not fulfilled by this example. Also I did not understand what "generalize the CHMW no-go result to include discrete symmetry" means and how it fits in the discussion. Of course there are known generalizations (in some sense) of CHMW no-go result to include discrete symmetry, surely the authors mean generalizations in a different direction, but it should be explained.

  • Minor but not so minor: there are -many- missing articles. I cannot possibly write all the instances here. Please consider re-reading the text and adding articles where needed.

  • To improve clarity, I would write that eq. (2.18) comes from setting $\delta\alpha = 0$ in (2.17).

  • At the beginning of section 2.3 "we would like to make sure" should be changed with "we would like to provide some checks" (of course one cannot be sure by looking at a few scaling dimensions).

  • Below equations (3.6) "one-dimensional family of minima" if I understand correctly the authors count dimension in the space of $\phi^2$, I find that a little bizarre, usually one counts dimensions in the space of $\phi^i$.

  • validity: -
  • significance: -
  • originality: -
  • clarity: -
  • formatting: -
  • grammar: -

Author:  Noam Chai  on 2022-04-14  [id 2385]

(in reply to Report 2 on 2022-03-19)
Category:
remark
answer to question

Dear Editor, We thank our referee for thoroughly reading our manuscript. We accounted for all of his/her comments and suggestions in the new version of the paper.

  • We accept referee's comment regarding the conformal manifold, and therefore we simply removed a confusing phrase in the introduction "to our knowledge there were no prior known examples of the non-SUSY finite N models admitting an exactly marginal deformation"

*We modified our phrasing above (2.21) and below (2.31) to stress the point raised by the referee. In particular, we cited recent references mentioned in the report. The derivation of the known result is preserved because it is short and makes our presentation seamless.

Above (2.21): "We begin with repeating the derivation of the anomalous dimensions at leading order of the conformal perturbation theory \cite{papers mentioned by the referee}."

Below (2.31) "To recapitulate, at leading order of the conformal perturbation theory the anomalous dimensions are given by the eigenvalues of the derivatives matrix ${\partial\beta_i\over\partial g_j}$ evaluated at the fixed point \cite{papers mentioned by the referee}."

*We rephrased the last paragraph on page 19. Indeed, the 2+1 dimensional model in [8] exhibits persistent symmetry breaking of discrete symmetry. However, this model is manifestly non-local. So, it is still an open question whether a local model with persistent symmetry breaking exists. Hence, locality is the desideratum in addition to UV completeness.

*We added two more citations for conformal perturbation theory - "Scaling and renormalization in statistical physics" by Cardi, and https://arxiv.org/abs/hep-th/9109041.

  • To improve clarity, we changed our wording above (2.18) as follows: "Corresponding fixed points are the intersections of (2.14) with the surface obtained by setting $\delta\alpha=0$ in (2.17), see Fig. 1, "

*At the beginning of section 2.3 "we would like to make sure" is replaced by "we would like to provide some checks"

*To avoid confusion, we paraphrased one of the sentences below (3.6) as follows "This defines a one-dimensional family of minima in the space of fields – a hyperbola in the $(\phi_1, \phi_2)$ plane."

We hope that our paper can now be published in SciPost. Sincerely,

Noam Chai, Anatoly Dymarsky, Mikhail Goykhman, Ritam Sinha, and Michael Smolkin

---

## Editorial Decision

resubmitted